# Diabetic Encephalopathy in a Preclinical Experimental Model of Type 1 Diabetes Mellitus: Observations in Adult Female Rat

**DOI:** 10.3390/ijms24021196

**Published:** 2023-01-07

**Authors:** Eva Falvo, Silvia Giatti, Silvia Diviccaro, Lucia Cioffi, Monika Herian, Paola Brivio, Francesca Calabrese, Donatella Caruso, Roberto Cosimo Melcangi

**Affiliations:** Dipartimento di Scienze Farmacologiche e Biomolecolari, Università degli Studi di Milano, 20133 Milan, Italy

**Keywords:** cognition, brain, sex-dimorphism, dihydroprogesterone, allopregnanolone, neuroinflammation, oxidative stress, mitochondrial functionality

## Abstract

Patients affected by diabetes mellitus (DM) show diabetic encephalopathy with an increased risk of cognitive deficits, dementia and Alzheimer’s disease, but the mechanisms are not fully explored. In the male animal models of DM, the development of cognitive impairment seems to be the result of the concomitance of different processes such as neuroinflammation, oxidative stress, mitochondrial dysfunction, and aberrant synaptogenesis. However, even if diabetic encephalopathy shows some sex-dimorphic features, no observations in female rats have been so far reported on these aspects. Therefore, in an experimental model of type 1 DM (T1DM), we explored the impact of one month of pathology on memory abilities by the novel object recognition test and on neuroinflammation, synaptogenesis and mitochondrial functionality. Moreover, given that steroids are involved in memory and learning, we also analysed their levels and receptors. We reported that memory dysfunction can be associated with different features in the female hippocampus and cerebral cortex. Indeed, in the hippocampus, we observed aberrant synaptogenesis and neuroinflammation but not mitochondrial dysfunction and oxidative stress, possibly due to the results of locally increased levels of progesterone metabolites (i.e., dihydroprogesterone and allopregnanolone). These observations suggest specific brain-area effects of T1DM since different alterations are observed in the cerebral cortex.

## 1. Introduction

Sustained hyperglycaemia, as in diabetes mellitus (DM), may induce severe complications in different organs, including damage to the central nervous system, leading to the so-called diabetic encephalopathy. Indeed, the association of DM with cognitive deficits and increased risk of dementia, stroke, cerebrovascular and Alzheimer’s disease, as well as psychiatric disorders (e.g., depression), has been well established [1,2,3,4,5,6,7,8,9]. Even if type 2 DM (T2DM) is more prevalent in the population, memory or cognitive deficits seem to be more relevant in type 1 DM (T1DM) patients. Indeed, T1DM patients present a 65% increased risk of dementia compared to the 37% of T2DM patients [10]. Diabetic encephalopathy in experimental models is characterized by swollen synaptic boutons and fragmentation of neurofilaments within the axons [11], swelling of axons and dendrites [12], alterations of myelin membranes [11] and in its lipid and protein components [13,14,15], leading to the impairment of axonal transport [16]. Streptozotocin (STZ)-treated rats (i.e., an experimental model of T1DM) show cognitive and memory impairment as well as alteration of hippocampal long-term potentiation [17,18]. These features have been proposed to be related to pre- and post-synaptic changes in the hippocampus [19]. In particular, the development of cognitive deficits in DM seems to be the result of the concomitance of different processes, such as oxidative stress [20,21,22], impairment of mitochondrial functionality [23,24,25], neuroinflammation [22,26,27,28,29,30,31] and aberrant synaptogenesis characterized by synaptic protein loss and synaptic structure impairments [18,27,32,33]. Another factor involved in the etiopathogenesis of diabetic encephalopathy may be the sex steroid environment. Sex steroids are synthesized in the peripheral glands (i.e., sex steroid hormones) as well as directly in the nervous system (i.e., neurosteroids) and are important physiological regulators of the nervous function [34,35,36,37]. Neuropathological conditions affect their levels, and consequently, different steroid molecules have been clearly demonstrated to exert neuroprotective effects [35,38,39]. In agreement, DM influences the levels of sex steroids in plasma due to dysfunction in the reproductive axis [40,41,42] as well as in the brain [43,44,45]. In particular, as demonstrated in the STZ model, the brain levels of important neuroactive steroids (i.e., the family of steroids including steroid hormones and neurosteroids), such as pregnenolone (PREG), progesterone (PROG) and testosterone (T) as well as of their metabolites (i.e., dihydroprogesterone, DHP, allopregnanolone, ALLO and isoallopregnanolone, ISOALLO, for PROG, and dihydrotestosterone, DHT and 5alpha-androstane-3alpha,17beta-diol, 3α-diol, for T) are decreased by three months of DM (long-term diabetes) [43,44,45]. Interestingly, the hippocampal levels of these steroids are already affected in male STZ rat after one month of pathology [46]. However, no observations on female diabetic rats have been reported so far. Exploring the steroid environment, as well as cognitive abilities and related mechanisms in female diabetic animals, could be extremely interesting because diabetic encephalopathy shows some sex-dimorphic features. For instance, young patients with T1DM perform poorly in school compared to healthy classmates, showing reduced performance and intelligence quotient. This decline was observed only in boys with diabetes and not in girls diagnosed before the age of six [47]. Moreover, as mentioned above, diabetic encephalopathy is associated with neurological decline [2,48,49,50,51] showing sex differences in terms of incidence, progression and severity [52,53,54,55,56,57,58,59].

We have here investigated the impact of one month of diabetes on (1) memory abilities, (2) neuroinflammation, (3) synaptogenesis, (4) oxidative stress, (5) mitochondrial functionality and (6) plasma and brain levels of neuroactive steroids in adult female animals. Specifically, memory function has been explored by the novel object recognition (NOR) test in STZ rats and controls on diestrous days. Neuroinflammation and synaptogenesis have been analysed in brain areas (i.e., hippocampus and cerebral cortex) of diabetic and control rats, measuring the mRNA levels of proteins involved in the inflammatory response, such as interleukin (IL)-6 and IL-1β, Toll-like receptor 4 (TLR4) and tumor necrosis factor-alpha (TNFα) by real time-PCR and levels of synaptophysin, synapsin and syntaxin by Western blot analysis. Oxidative stress has been analysed in the plasma, hippocampus, and cerebral cortex by thiobarbituric acid reactive substances (TBARS) assay, as an index of reactive oxygen species production, and by protein levels of superoxide dismutase 2 (SOD2) in the hippocampus and cerebral cortex. Mitochondrial functionality has been studied in the hippocampus and cerebral cortex by assessment of respiratory chain complexes functional subunits (OXPHOS). Finally, in the same brain areas and in plasma, the levels of steroids, such as PREG, dehydroepiandrosterone (DHEA), PROG and its metabolites, T and its metabolites, and 17β-estradiol (17β-E), were assessed by liquid chromatography-tandem mass spectrometry (LC-MS/MS). Based on the results obtained on steroid levels present in the hippocampus, the gene expression of the PROG receptor (PR) and of different subunits of the GABA-A receptors have been assessed in this brain area.

## 2. Results

### 2.1. Blood Glucose Levels and Body Weight of Female T1DM and Control Rats

As expected, the STZ administration induced typical diabetic manifestations. All animals treated with STZ had increased levels of glucose in plasma (above 300 mg/dL), however, they did not lose body weight compared to non-diabetic animals (Table 1).

### 2.2. Novel Object Recognition Performance in Female T1DM and in Control Rats

Novel object recognition (NOR) test was used to evaluate memory abilities in an experimental model of T1DM versus control. As shown in Figure 1, one month of DM induced by STZ treatment in female rats induced, at the diestrous phase, a significant decrease in the NOR index compared to the control. Thus, T1DM significantly affects memory function in female animals.

### 2.3. Neuroinflammation Markers and Synaptic Proteins in the Hippocampus and Cerebral Cortex of Female T1DM and Control Rats

To assess the molecular modifications producing impaired memory function, we evaluated in the hippocampus neuroinflammation and aberrant synaptogenesis, both of which are important mechanisms involved in diabetic encephalopathy and memory impairment. Therefore, we first analysed the gene expression of important cytokines in our experimental model. As shown in Figure 2, our data reveal that the relative mRNA expression of IL-1β (panel A) and IL-6 (panel B) was significantly increased by DM in the hippocampus of STZ female rats. No statistical difference was observed in the expression of two other inflammatory factors considered, such as TLR4 and TNFα (panel C,D). The expression of synaptic proteins was also affected by one month of DM in the same brain region of STZ female rats. Indeed, as reported in Figure 2, levels of synaptophysin (panel E), synapsin (panel F) and syntaxin (panel G) were significantly decreased in diabetic animals compared to the control group.

These effects seem to be specific to the hippocampus. Indeed, analyses performed in the cerebral cortex (Figure 3) showed that the relative mRNA expression of IL-1β (panel A) and IL-6 (panel B), as well as in the expression of the synaptic protein synaptophysin (panel E) and syntaxin (panel G) were unaffected by DM. In contrast to what was observed in the hippocampus (Figure 2 panel C,D), the relative mRNA expression of TLR4 (panel C) and TNFα (panel D) in the cerebral cortex was significantly increased by short-term diabetes. In addition, unlike what was reported in the hippocampus (Figure 2, panel F), the expression of synapsin in the cerebral cortex of STZ -female animals (Figure 3, panel F) was significantly increased by pathology.

### 2.4. Oxidative Stress and Mitochondrial Functionality in Female T1DM and Control Rats

As an index of reactive oxygen species production, we assessed the levels of TBARS in our experimental model. As reported in Figure 4, the TBARS levels were significantly increased in the plasma (panel C) and in the cerebral cortex (panel B) but not in the hippocampus (panel A) of STZ female animals, indicating oxidative stress in the periphery and in the cerebral cortex, and consequently suggesting a specific effect depending on the brain area considered. As reported in panel D, the protein levels of SOD2 were upregulated in the hippocampus but significantly decreased in the cerebral cortex of STZ female animals by diabetes (panel E).

Mitochondrial functionality has been assessed by detecting OXPHOS levels. As reported in Figure 4 panel F, the protein content of different subunits belonging to respiratory chain NDUFA8B (complex I), SDHB (complex II), UQCRC2 (complex III), mt-COX2 (complex IV) and ATP5A (complex V) was unmodified in the hippocampus. On the contrary, except for complex II and complex III, the complexes I, IV and V were significantly decreased in the cerebral cortex of female T1DM rats (panel G).

### 2.5. Neuroactive Steroid Levels in the Hippocampus, the Cerebral Cortex and Plasma of Female T1DM and Control Rats

The levels of neuroactive steroids were assessed by LC-MS/MS and reported in Figure 5. As reported in panel A, a significant increase in the PROG metabolites, DHP and ALLO, was observed in the hippocampus of the STZ female rats compared with the controls. The levels of the other steroids assessed were not significantly modified by pathology. The effect of DM was different in the cerebral cortex (Figure 5 panel B). Indeed, only a significant increase in the levels of another PROG metabolite (i.e., ISOALLO) was observed. The assessment of steroids in plasma (Figure 5 panel C) showed a different pattern compared with that observed in the brain areas here considered. Indeed, the levels of PREG and of its direct metabolite, PROG, as well as those of PROG metabolites, DHP and ALLO, were reduced in STZ female rats compared to control animals.

### 2.6. Gene Expression of Progesterone Receptor and GABA-A Receptor Subunits in the Hippocampus of T1DM Female and Control Rats

DHP and ALLO interact with different receptors (i.e., DHP with PR and ALLO with GABA-A receptor). Therefore, we have evaluated whether changes in the levels of DHP and ALLO observed in the hippocampus of diabetic female rats were associated with changes in the gene expression levels of their receptors. As shown in Figure 6, panel A, the gene expression levels of PR were similar in STZ and control animals. The assessment of different GABA-A receptor subunits showed that the levels of the γ2 subunit, but not those of α1, α3, α5, β2 and δ subunits, were significantly decreased in the hippocampus of STZ vs CTRL female rats (panel B).

## 3. Discussion

Data herein reported indicate that one month of T1DM induced by STZ in female rats is able to decrease memory abilities, evaluated by the NOR test and that, in the hippocampus, this effect is associated with neuroinflammation (i.e., an increase in the levels of IL-1β and IL-6) and aberrant synaptogenesis (i.e., decrease in the levels of synaptophysin, synapsin and syntaxin) but not with oxidative stress, measured by TBARS levels and as confirmed by the lack of alteration in the mitochondrial functionality, measured by OXPHOS levels. These effects seem to be brain area specific. Indeed, we also analysed molecular parameters in the cerebral cortex, a brain region connected with the hippocampus that contributes to the recognition of memory for objects [60]. As reported here, IL-1β and IL-6 levels, as well as those of synaptophysin and syntaxin, are unaffected in the female cerebral cortex. In contrast, an increase in TNFα, TLR4 and in synapsin levels, along with the oxidative stress and altered mitochondrial functionality have been observed in this brain area. These data suggest that, in the cerebral cortex, different mechanisms in comparison with the hippocampus contribute to the presence of memory deficits.

The difference in the results obtained in the hippocampus and cerebral cortex could be related to the different effects of T1DM on the neuroactive steroid levels present in these two brain areas. Indeed, we reported that one month of T1DM induced an increase in DHP and ALLO levels in the hippocampus, whereas in the cerebral cortex only an increase in ISOALLO levels was observed. These T1DM effects are specific for these brain areas since, in plasma, levels of DHP and ALLO were decreased and those of ISOALLO were unaffected. Additionally, a decrease in the plasma levels of PREG and PROG was reported. Thus, the neuroactive steroid levels in these two brain areas are differently affected by the one month of pathology and the changes occurring in these brain areas do not reflect the effect of T1DM in the plasma levels. Brain region specificity and differences between brain and plasma steroid levels have also been demonstrated in various physiological and pathological conditions [39,61]. 

Therefore, the increase in DHP and ALLO levels in the hippocampus but not in the cerebral cortex and plasma could be interpreted as a possible protective effect of these molecules. Indeed, these two progesterone metabolites act as neuroprotective agents [62,63,64,65,66] by interaction with PR in the case of the DHP or GABA-A receptor in the case of ALLO [34,67,68,69,70]. As reported here, PR mRNA levels are not affected in the hippocampus of female T1DM rats; however, the mRNA levels of the γ2 subunit of the GABA-A receptor were significantly decreased. Indeed, as previously reported, ALLO treatment is able to decrease the gene expression of this GABA-A subunit [71,72]. Therefore, a possible hypothesis may link the increase in ALLO levels observed in the hippocampus with the absence of oxidative stress and alteration in mitochondrial functionality.

Indeed, as reported in several neuropathological experimental models, ALLO treatment reduces oxidative stress and protects mitochondria [73,74,75]. For instance, in vitro the literature data supports the idea that ALLO protects from oxidative stress by increasing SOD2 activity or expression [76,77]. Similarly, in an experimental model of status epilepticus, ALLO treatment induced higher SOD2 expression. In turn, it reduces cell death, DNA fragmentation, oxidative DNA damage and ROS production in the hippocampus [74]. In addition, ALLO is able to increase the mitochondrial oxygen consumption rate and ATP production in the SH-SY5Y cells. Furthermore, after exposure to H_2_O_2_ alone or in combination with amyloid beta (Aβ) overproduction, ALLO was able to increase the reduced levels of ATP and decrease oxidative stress [75]. 

A further support to the hypothesis that in the hippocampus, ALLO may be protective against oxidative stress and alteration in mitochondrial functionality is the finding that decreased OXPHOS levels and increased TBARS levels occurred in the cerebral cortex where ALLO levels are unaffected. 

To our knowledge, the only data available about the impact of short-term experimental T1DM on memory function have been obtained in male animals. The literature data also indicate that, in agreement with what is herein reported, in male animals, the memory impairment induced by STZ injection is associated with neuroinflammatory patterns and alterations in synaptic proteins in the hippocampus [27,78,79,80]. On the other side, oxidative stress [79,81,82,83] and altered mitochondrial functionality were reported in the male hippocampus [46,84,85], suggesting different mechanisms in the two sexes possibly related to the decrease in memory abilities. This sex–brain difference seems to be specific to the hippocampus. Indeed, similarly to what was reported in the male T1DM rats, oxidative stress [20,21,22,86] and altered mitochondrial functionality [86] were herein observed in the cerebral cortex of female T1DM rats. However, some small sex differences may also occur. For instance, the TBARS levels were increased in the female cerebral cortex but not in males, probably because in males SOD2 is upregulated [86]. Additionally, a significant decrease in complexes I, IV and V was herein observed in females, while in males only a decrease in complex IV was reported [86]. In terms of oxidative stress, the sex dimorphism reported in the hippocampus was not present in the periphery. Indeed, TBARS plasma levels increase both in females and males [46,86]. This may be ascribed to different mechanisms involved in the oxidative stress occurrence in the brain and in blood circulation. 

As we previously reported, the levels of neuroactive steroids are affected in the brain regions and the plasma of T1DM male rats. However, the impact of DM on these molecules is different in the two sexes. Indeed, compared to female animals, in the male hippocampus [46] and cerebral cortex [86], we reported a decrease in the levels of PREG, PROG, ALLO, T, DHT and 3α-diol. In addition, in the male hippocampus, a decrease in the levels of ISOALLO was also reported [46]. Like in the case of female diabetic animals, also changes in plasma levels of male diabetic animals did not exactly reflect those observed in brain regions and show sexual dimorphism. Indeed, in the plasma of male diabetic animals, a decrease in the levels of ISOALLO, T and 3α-diol was observed [46,86]. Sex brain region specificity, as well as sex differences between the brain and plasma steroid levels have also been demonstrated in physiological and pathological conditions [35,87,88].

In conclusion, data reported suggest that in the female rat hippocampus affected by T1DM, memory dysfunction can be associated with aberrant synaptic function and neuroinflammation. However, mitochondrial functionality and oxidative stress are not affected in this brain area, possibly due to the results of locally increased levels of DHP and ALLO. Additionally, these issues are specific to the hippocampus since the cerebral cortex does not present a similar increase in steroid levels and, consequently, shows increased neuroinflammation and oxidative stress, possibly due to dysfunctional mitochondria. Overall, these data indicate that in the female brain, there is an attempt to counteract the effects of T1DM, specifically in the hippocampus, which is probably not effective and thus produces memory deficits. However, one limitation of this study is the use of the NOR as a unique test to evaluate memory dysfunction. Thus, future experiments with tests exploring different memory or cognitive domains may reveal greater protection exerted by neuroactive steroids in the female brain. Additionally, it will also be important to evaluate the possible role of neuroactive steroids in the cognitive decline observed in T2DM.

## 4. Materials and Methods

### 4.1. Animals

Female Sprague-Dawley rats (150–175 g at arrival, Charles Rivers Laboratories, Lecco) were used. Animals were housed in the animal care facility of the Dipartimento di Scienze Farmacologiche e Biomolecolari (Università degli Studi di Milano, Milan, Italy). All animals were kept in Individually Ventilated Cages (IVC), with food and tap water available ad libitum and under controlled temperature (21 ± 4 °C), humidity (40–60%), room ventilation (12.5 air changes per h) and light cycles (12 h light/dark cycle; on 7 A.M./off 7 P.M.).

The rats were allowed to acclimate to new environment for 7 days before being randomly assigned to one of the experimental groups described below. Animal care and procedures were approved by our institutional animal use and care committee and are in line with Institutional guidelines that are following national (D.L. No. 26, 4 March 2014, G.U. No. 61 14 March 2014) and international laws and policies (EEC Council Directive 2010/63, 22 September 2010: Guide for the Care and Use of Laboratory Animals, United States National Research Council, 2011).

### 4.2. Diabetic Induction and Characterisation

Female animals were randomly divided into the following two different experimental groups: (i) non-diabetic control animals (CTRL); (ii) streptozotocin diabetic animals (STZ). To obtain diabetic condition, rats were injected with a single intraperitoneal of freshly prepared streptozotocin (60 mg/kg body weight; Sigma-Aldrich, Milan, Italy) in citrate buffer (0.09 M pH 4.8) as previously described [44]. Non-diabetic control animals received injections of citrate buffer alone. After 48 h, diabetes was confirmed by tail vein blood glucose measurement using a commercial glucometer (Contour next, Ascensia Diabetes Care Italy, Milan, Italy) and only the rats with feeding blood glucose above 300 mg/dL were classified diabetic. Body weight was monitored every week. After one month from the determination of hyperglycaemic status, both CTRL and STZ animals were sacrificed in diestrus and the hippocampus, the cerebral cortex and plasma were collected and stored at −80 °C until analysed. Blood samples were collected in tubes with EDTA 0.25 M and centrifugated at 2500× *g* for 15 min at 4 °C to obtain plasma.

### 4.3. Estrus Cycle Analysis

Estrus cycle patterns were evaluated using vaginal smear method. Vaginal smears were collected once a day, at the same time (9.00 a.m.–10.00 a.m.) for 7 consecutive days. The smear slides were analysed under light microscopy to determinate the cell types present and then the following four different phases: proestrus, estrus, metestrus and diestrus.

### 4.4. Novel Object Recognition (NOR) Test

After one month of diabetes, animals were tested in non-transparent open field made of plexiglas (60 × 60 × 60 cm) during the diestrous phase. After 1 h of adaptation session to the room of test, the animals were allowed to explore two identical objects in the open field for 5 min (trial session-encoding phase). During this session, all the subjects completed a minimum of 15 s of time required of exploration. Then rats were returned to their home cages for one hour (retention phase) and one of the objects presented previously was replaced by a novel object. Then, rats were returned to the open field for 5 min (testing session) and the duration of exploration of each object, the familiar and the novel one, (i.e., sitting near the objects, sniffing, or touching it) was manually measured during the 5 min test by two independent observers, blind to the experimental design. The task was performed in an isolated room under low light conditions, in the absence of direct overhead lighting. The NOR index was calculated according to the following formula: time of novel object exploration divided by time of novel plus familiar object exploration, multiplied by 100.

### 4.5. Liquid Chromatography–Tandem Mass Spectrometry Analysis (LC-MS/MS)

For the quantitative analysis of neuroactive steroids (namely, PREG, PROG, DHP, ALLO, ISOPREG, DHEA, T, DHT, 3α-diol and 17β-E), the hippocampus (50 mg), cerebral cortex (85 mg) and plasma (300 mL) were extracted and purified as we previously described [89]. The 17β-Estradiol-2,3,4-^13^C_3_ (^13^C_3_-17β-E) (2 ng/sample), progesterone-2,3,4,20,25-^13^C_5_ (^13^C_5_– PROG) (0.4 ng/sample) and pregnenolone-20,21-^13^C_2_-16,16 D_2_ (^13^C_2_D_2_– PREG) (10 ng/sample), were used as internal standards. Neuroactive steroid levels were assessed based on calibration curves freshly prepared and extracted [89].

The samples were spiked with labelled internal standards. Hippocampus and cerebral cortex samples were homogenized in MeOH/acetic acid (99:1 *v*/*v*) using a tissue lyser (Qiagen, Milan, Italy) and plasma samples were diluted in ACN. After an overnight extraction at 4 °C, the organic phase of each sample was evaporated to dryness, resuspended in MeOH/H_2_O 1:9 (*v*/*v*) and purified using C18 SPE cartridges (HyperSep C18 SPE Columns 500 mg 3 mL; Microcolumn, Milano, Italy). The steroids were eluted in MeOH, concentrated and transferred in autosampler vials before LC-MS/MS analysis. The analysis was conducted by liquid chromatography (LC) supplied by Surveyor LC Pump Plus and Surveyor Autosampler Plus (Thermo Fisher Scientific, Waltham, MA, USA) connected with a linear ion trap—mass spectrometer LTQ (Thermo Fisher Scientific, MA, USA), operated in positive atmospheric pressure chemical ionization (APCI+) mode. The chromatographic separation was achieved with a Hypersil Gold column C18 (100 × 2.1 mm, 3 µm; Thermo Fisher Scientific, MA, USA) maintained at 40 °C. The mobile phases consisted of 0.1% formic acid in H_2_O (phase A) and 0.1% formic acid in MeOH (phase B). Gradient elution was as follows: 0–1.50 min 70% A; 1.50–2.00 min 55% A; 2.00–3.00 min 55% A, 3.00–35.00 min linear gradient to 36% A; 35.00–40.00 min 25% A; 41.00–45.00 min 1% A; 45.00–45.20 min 70% A and 45.40–55.00 min equilibrate with 70% A. The 25 μL sample was injected at a flow rate of 300 µL/min. The divert valve was set at 0–8 min to waste, 8–45 min to source and 45–55 min to waste. The injector needle was washed with MeOH/H_2_O 1:1 (*v*/*v*). LC-MS/MS data were acquired and processed using software Excalibur^®^ release 2.0 SR2 (Thermo Fisher Scientific, MA, USA).

### 4.6. Real-Time Polymerase Chain Reaction

RNA was extracted from snap-frozen hippocampus and cerebral cortex using Directzol ™MiniPrep kit (Zymo Research, Irvine, CA, USA) following manufacturing protocol. The quantification of RNA was performed by NanoDrop™ 2000 (ThermoFisher scientific, Milano, Italy). Gene expression was assessed by TaqMan quantitative real-time PCR using a CFX96 real-time system (Bio-Rad Laboratories, Segrate, Italy). Samples run in 96-well formats in duplicate as multiplexed reactions with a normalizing internal control, 36B4 (Eurofins MWG-Operon, Milano, Italy) using the iTaq™ Universal Probes One-Step Kit (Bio-Rad, Segrate, Italy). Specific TaqMan MGB probes and primers sequences were purchased at Life Technologies Italia (Monza, Italy) or at Eurofins MWG Operon (Milano, Italy). Eurofins MWG Operon: IL-6 fwd: AAGCCAGAGTCATTCAGAGC rev: GTCCTTAGCCACTCCTTCTG; IL-1β fwd: TGCAGGCTTCGAGATGAAC rev: GGGATTTTGTCGTTGCTTGTC; TLR4 fwd: CATGACATCCCTTATTCAACCAAG rev: GCCATGCCTTGTCTTCAATTG; TNFα fwd: CTTCTCATTCCTGCTCGTGG rev: TGATCTGAGTGTGAGGGTCTG; GABA-A α1 fwd: GAGAGTCAGTACCAGCAAGAAC rev: AGAACACGAAGGCATAGCAC; GABA-A α3 fwd: TTCACTAGAATCTTGGATCGGC rev: TCTGACACAGGGCCAAAAC; GABA-A α5 fwd: GATCGGGTACTTTGTCATCCAG rev: TGATGCTGAGGGTTGTCATG; GABA-A β2 fwd: CTGGATGAACAAAACTGCACG rev: ACAATGGAGAACTGAGGAAGC.

Whereas specific primers and probe mix for GABA-A δ (Rn01517017_g1), GABA-A γ2 (Rn00788325_m1) and PR (Rn00575662_m1) were purchased from Life Technologies Italia (Monza, Italy). The mean control value within a single experiment was set to 1 and all the other values were expressed as the ratio with the controls.

### 4.7. Western Blotting

For Western blotting, the hippocampus and cerebral cortex were homogenized using the Tissue Lyser (Qiagen, Italy) in a cold lysis buffer (PBS without Ca^2+^ and Mg^2+^, EDTA 0.5 M pH 8, Igepal) supplemented with a protease cocktail inhibitor (Life Technologies, Carlsbad, CA, USA). Homogenates were centrifugated at 2000 rpm for 5 min at 4 °C to remove particulate matter. The protein content of lysates was quantified using a Bradford Assay (Bio-Rad, Segrate, Italy). Then, samples containing equal amounts of protein were heated to 100 °C for 5 min, or to 37 °C for 5 min in case of OXPHOS, and then run on polyacrylamide gel and then transferred to nitrocellulose membranes. For immunoblot detection, membranes were cut and then blocked on an orbital shaker for 1 h at room temperature in 10% non-fat dry milk or 5% bovine serum albumin (BSA). Successively, each membrane was exposed to primary antibodies, which are listed in Table 2. Primary antibody of Synaptophysin and SOD2 was used at 1:1000 dilution in 5% BSA, while primary antibodies of synapsin and syntaxin and OXPHOS in PBS-T 2.5% non-fat dry milk, while for GAPDH and VDAC, as the protein housekeeping, primary antibodies were used at 1:10,000 and 1:2000 dilution in PBS-T 2.5% non-fat dry milk and BSA 5%, respectively. After overnight incubation at 4 °C and extensive washing, the membranes were incubated with an anti-rabbit or anti-mouse horseradish peroxidase-conjugated secondary antibody, according to the primary antibody (see Table 2). After washing, the protein bands were detected on membranes using the ECL method (Bio-Rad, Segrate, Italy). ECL signals were acquired with a ChemiDocTM XRS+ system (Bio-Rad, Segrate, Italy) and analysed with Image Lab^TM^ software version 5.2.1 (Bio-Rad, Segrate, Italy). The mean control value within a single experiment was set to 100 and all the other values were expressed as a percentage.

### 4.8. Thiobarbituric Acid-Reactive Substance

Tissue and plasmatic thiobarbituric acid-reactive substances (TBARS) were determined as an index of reactive oxygen species (ROS) production. We used the formation of TBARS during an acid-heating reaction, which is widely adopted as a sensitive method for the measurement of lipid peroxidation, as previously described [90], with modifications. Briefly, 10 μg of hippocampus or cerebral cortex, were homogenized in 400 μL of lysis buffer (TrisHCl 0.1 M, pH 7.4; EDTA 1.34 mM; glutathione 0.65 mM) using a TissueLyser II (Qiagen, Hilden, Germany). Then, 100 μL of homogenate was mixed with 600 μL of phosphoric acid 1% and 200 μL of TBA 0.6%. Samples were then incubated at 95 °C for 1 h and then cooled to RT, extracted with 1 mL of n-butanol and then centrifuged at 3000 rpm at 4 °C for 20 min. For the analysis of plasma, we used 100 μL of plasma stored at −20 °C with a mixture of antioxidant reagent, EDTA 1.34 mM and glutathione 0.65 mM, then the protocol was the same as for the tissues except the homogenization phase. The supernatant was measured fluorometrically at an excitation wavelength of 532 nm and an emission wavelength of 553 nm. Quantification was performed using the standard curve prepared with malondialdehyde following similar conditions.

### 4.9. Statistical Analysis

Behavioural, LC-MS/MS, real-time PCR, TBARS assay and Western blot data were analysed by unpaired two-tailed Student’s *t*-test, after checking normal distribution with the Kolmogorov-Smirnov test. Significance for all tests was assumed for *p* < 0.05. Data are presented as means standard error (SEM). Analyses were performed using Prism, version 7.0a (GraphPad Software Inc., San Diego, CA, USA).

## Figures and Tables

**Figure 1 ijms-24-01196-f001:**
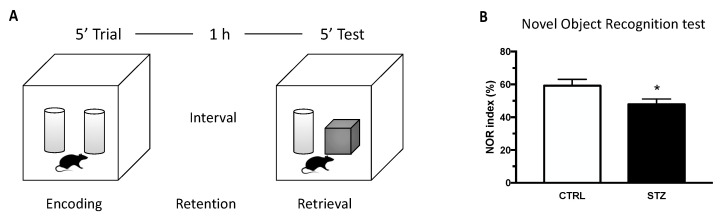
Effect of one month of diabetes on novel object recognition (NOR) performance. NOR test was carried in non-diabetic control (CTRL; n = 7) and diabetic (STZ; n = 7) female animals. (Panel **A**) schematic picture of the novel object recognition test. (Panel **B**) NOR index evaluated at the end of the experiment (4 weeks). The columns represent the mean ± SEM. Statistical analysis was performed by unpaired two-tailed Student’s *t*-test. * *p* < 0.05.

**Figure 2 ijms-24-01196-f002:**
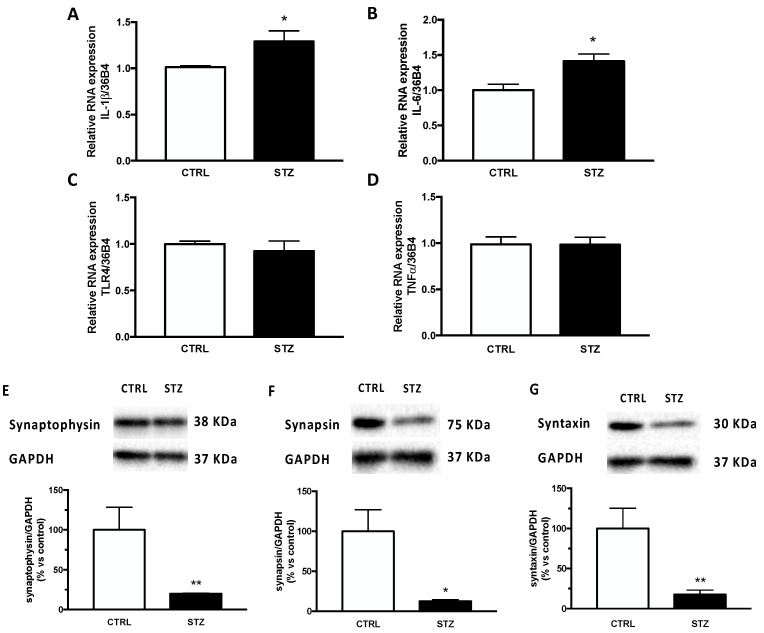
Effect of one month of diabetes on the gene expression of neuroinflammation markers and protein content of synaptic proteins in the hippocampus of female rats. IL-1β (panel **A**), IL-6 (panel **B**), TLR4 (panel **C**) and TNFα (panel **D**). The columns represent the mean ± SEM after normalization with 36B4 in non-diabetic control (CTRL; n = 7) and diabetic (STZ; n = 7) female animals. Synaptophysin (panel **E**), synapsin (panel **F**) and syntaxin (panel **G**). The protein levels were detected by Western blotting in non-diabetic control (CTRL; n = 7) and diabetic (STZ; n = 7) female animals. Above each histogram a representative blot is shown. The columns represent the mean ± SEM after normalization with GAPDH. Statistical analysis was performed by unpaired two-tailed Student’s *t*-test. * *p* < 0.05 and ** *p*< 0.01.

**Figure 3 ijms-24-01196-f003:**
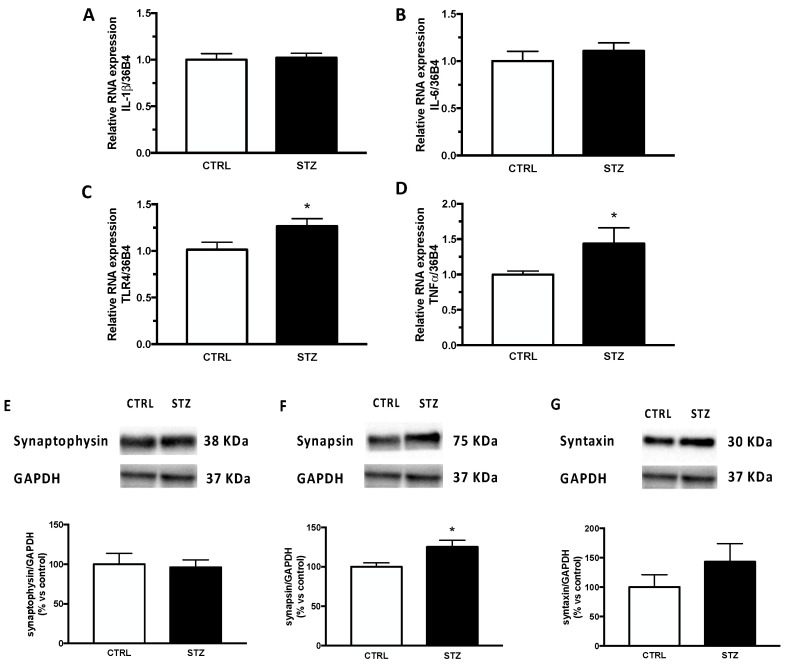
Effect of one month of diabetes on the gene expression of neuroinflammation markers and on protein content of synaptic proteins in the cerebral cortex of female rats. IL-1β (panel **A**), IL-6 (panel **B**), TLR4 (panel **C**) and TNFα (panel **D**). The columns represent the mean ± SEM after normalization with 36B4 in non-diabetic control (CTRL; n = 7) and diabetic (STZ; n = 7) female animals. Synaptophysin (panel **E**), synapsin (panel **F**) and syntaxin (panel **G**). The protein levels were detected by Western blotting in non-diabetic control (CTRL; n = 7) and diabetic (STZ; n = 7) female animals. Above each histogram a representative blot is shown. The columns represent the mean ± SEM after normalization with GAPDH. Statistical analysis was performed by unpaired two-tailed Student’s *t*-test. * *p* < 0.05.

**Figure 4 ijms-24-01196-f004:**
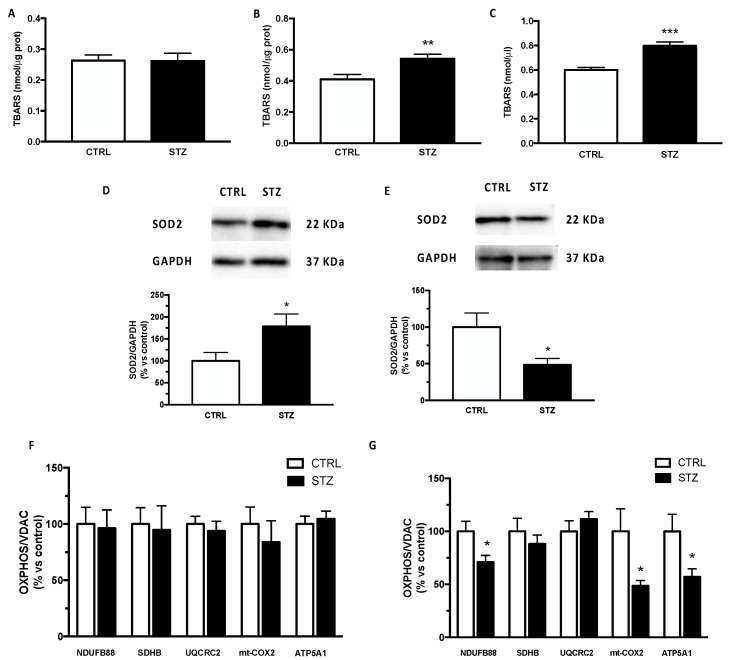
Effect of one month of diabetes on molecules involved in oxidative stress and mitochondrial functionality in the hippocampus, cerebral cortex and plasma of female rats. Thiobarbituric acid reactive substance (TBARS) in hippocampus (panel **A**), in cerebral cortex (panel **B**) and in plasma (panel **C**). The columns represent the mean ± SEM. The protein levels of superoxide dismutase 2 (SOD2) were detected by Western blotting in hippocampus (panel **D**) and cerebral cortex (panel **E**) of non-diabetic control (CTRL; n = 7) and diabetic (STZ; n = 7) female animals. Above each histogram a representative blot is shown. The columns represent the mean ± SEM after normalization with GAPDH. Protein contents of NDUFA8B (complex I), SDHB (complex II), UQCRC2 (complex III), mt-COX2 (complex IV) and ATP5A1 (complex V) were detected by Western blotting in hippocampus (panel **F**) and cerebral cortex (panel **G**) of non-diabetic control (CTRL) and diabetic (STZ) female rats. The columns represent the mean ± SEM after normalization with VDAC. Statistical analysis was performed by unpaired two-tailed Student’s *t*-test. * *p* < 0.05 ** *p* < 0.01. *** *p* < 0.001.

**Figure 5 ijms-24-01196-f005:**
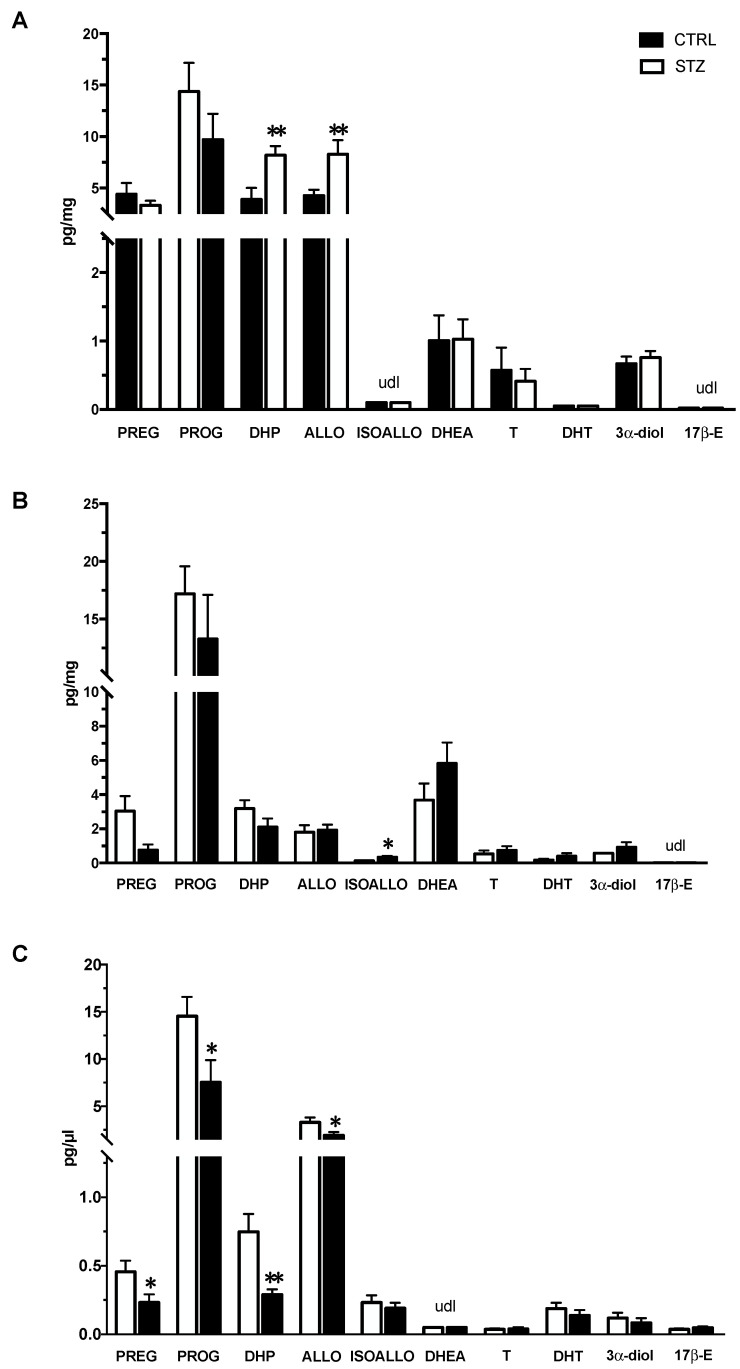
Effect of one month of diabetes on the levels of neuroactive steroids in the hippocampus, cerebral cortex and plasma of female non-diabetic control (CTRL; n = 7) and diabetic (STZ; n = 7) rats. Levels of neuroactive steroids in hippocampus (panel **A**), cerebral cortex (panel **B**) and in plasma (panel **C**). Data are expressed as pg/mg ± SEM in case of brain areas, and pg/μL ± SEM in case of plasma. u.d.l. = under detection limit. Detection limits were 0.02 pg/mg or pg/μL for testosterone (T) and 17β-Estradiol (17β-E), 0.05 pg/mg or pg/μL for pregnenolone (PREG), progesterone (PROG), 5α-androstane-3α,17β-diol (3α-diol), dehydroepiandrosterone (DHEA), dihydrotestosterone (DHT); 0.1 pg/mg or pg/μL for allopregnanolone (ALLO) and isoallopregnanolone (ISOALLO); 0.25 pg/mg or pg/μL for dihydroprogesterone (DHP). Statistical analysis was performed by unpaired two-tailed Student’s *t*-test. * *p* < 0.05. ** *p* < 0.01.

**Figure 6 ijms-24-01196-f006:**
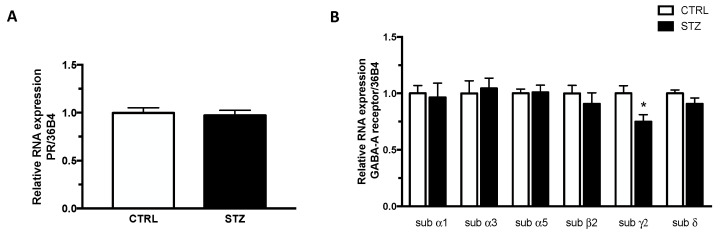
Effect of one month of diabetes on the gene expression of progesterone receptor (PR; panel **A**) and GABA-A receptor subunits, α1, α3, α5, β2, γ2, δ (panel **B**) in the hippocampus of female rats. The columns represent the mean ± SEM after normalization with 36B4 in non-diabetic control (CTRL; n = 7) and diabetic (STZ; n = 7) female animals. Statistical analysis was performed by unpaired two-tailed Student’s *t*-test. * *p* < 0.05.

**Table 1 ijms-24-01196-t001:** Body weight and glucose of non-diabetic (CTRL) and diabetic (STZ) rats.

Animal	Body Weight at Sacrifice (g)	Blood Glucose at Sacrifice (mg/dL)
CTRL	263.00 ± 5.62	115.90 ± 3.60
STZ	240.80 ± 6.55	533.40 ± 17.92 ***

Data are expressed as mean ± SEM. CTRL (n = 7) and STZ (n = 7). Statistical analysis was performed by unpaired two-tailed Student’s *t*-test. *** *p* < 0.001.

**Table 2 ijms-24-01196-t002:** List of primary antibodies.

Antibody	Code	Host
Synaptophysin	Cell signaling—5461S	Rabbit
Synapsin	Synaptic systems—106001	Mouse
Syntaxin	Synaptic systems—110011	Mouse
OXPHOS	ABCAM—AB10413	Mouse
SOD2	Sigma—PA001814	Rabbit
GAPDH	Santa Cruz—SC_25778	Rabbit
VDAC	ABCAM—AB15895	Rabbit

## Data Availability

Data are available on request.

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
