# Peer review of "Diabetic Encephalopathy in a Preclinical Experimental Model of Type 1 Diabetes Mellitus: Observations in Adult Female Rat"

_ijms, 2023, doi:10.3390/ijms24021196_

Round 1

Reviewer 1 Report

The aim of this manuscript is to study the possible mechanisms of diabetic encephalopathy in female rat model of T1DM. The author found that after 1 month of drug induced T1DM, female rat developed impaired cognitive deficits. They also found that this impaired memory ability was associated with abnormal hippocampus neuroinflammation and synaptogenesis, but not related to mitochondrial dysfunction and oxidative. No significant change in cerebral cortex was found in the study model.

1. A1c level, OGTT test, body weight after 1 months of drug-induce T1DM should be measured in both groups and present. 

2. Please fix the label of Fig 2B

3. In Fig 2 and Fig 3, it would be better to normalized CTRL gene expression to 1, which is easy for reader.

4. Please fix the label of Fig 4B

5. There is significant increased in TGF-alpha (Fig 3D) and more oxidative stress (Fig. 4B) in cerebral cortex, which may indicate the involvement of cerebral cortex. Please discuss and adjust the abstract accordingly. 

Author Response

We want to thank the Reviewers for his/her pertinent and useful comments that have clearly improved our manuscript. We have considered all their points of concern as detailed below.

Comment: The aim of this manuscript is to study the possible mechanisms of diabetic encephalopathy in female rat model of T1DM. The author found that after 1 month of drug induced T1DM, female rat developed impaired cognitive deficits. They also found that this impaired memory ability was associated with abnormal hippocampus neuroinflammation and synaptogenesis, but not related to mitochondrial dysfunction and oxidative. No significant change in cerebral cortex was found in the study model.

  1. A1c level, OGTT test, body weight after 1 months of drug-induce T1DM should be measured in both groups and present.

Answer: We have now added a table (new table 1) reporting the body weight and the glycemia of animals at the time of sacrifice, and modified accordingly the text. Unfortunately, we do not have other animals to perform OGTT and A1c tests. Indeed, the authorization to perform this study expired in these days, so at this time we cannot perform further analyses on this experimental model. Anyway, we wish to highlight that our experimental model is well confirmed as demonstrated by our previous observations (see for instance, doi: 10.1186/s13293-018-0164-z; doi: 10.1016/j.jsbmb.2017.11.009; doi: 10.1016/j.jsbmb.2016.11.019).

  1. Please fix the label of Fig 2B

 Answer: We did.

  1. In Fig 2 and Fig 3, it would be better to normalized CTRL gene expression to 1, which is easy for reader.

Answer: We applied this modification as requested and we extended the modification to Figure 6. We modify the methods accordingly.

  1. Please fix the label of Fig 4B

  Answer: We did.

  1. There is significant increased in TGF-alpha (Fig 3D) and more oxidative stress (Fig. 4B) in cerebral cortex, which may indicate the involvement of cerebral cortex. Please discuss and adjust the abstract accordingly.

        Answer: Please note that is TNF-alpha and not TGF-alpha. Anyway, we agree with the point because an involvement of cerebral cortex also occurs. We have now extended this aspect in the text.

Reviewer 2 Report

Falvo et al. presented diabetic encephalopathy in a preclinical experimental female rat model of type 1 diabetes mellitus (T1DM). This is an interesting paper, suggesting that 

in the female rat brain, memory dysfunction can be associated with aberrant synaptogenesis and neuroinflammation in the hippocampus but not with mitochondrial dysfunction and oxidative stress, possibly due to the results of locally increased levels of progesterone metabolites. This would be an important study focusing on the female brain, which is often neglected in animal experiments. There are, however, some issues to be addressed to further improve the manuscript.

1.     Why did the authors just focus on T1DM, not on T2DM? Because more than 90% of DM is type 2, the data of T2DM should be more useful for clinical consideration. Although streptozocin (STZ)-treated model is very popular, the reliable rat model of T2DM should be commercially available.

2.     As to the evaluation of aberrant synaptogenesis, the authors only analyzed the expression level of synaptic proteins such as synaptophysin, synapsin and syntaxin. The morphological changes of synapse should be required. 

3.     Although the authors used whole hippocampus and cerebral cortex, it is too rough.

For the hippocampus, it should be dorsal or ventral, and for the cortex, the region should be more restricted. 

Author Response

We want to thank the Reviewers for his/her pertinent and useful comments that have clearly improved our manuscript. We have considered all their points of concern as detailed below.

Comment: Falvo et al. presented diabetic encephalopathy in a preclinical experimental female rat model of type 1 diabetes mellitus (T1DM). This is an interesting paper, suggesting that in the female rat brain, memory dysfunction can be associated with aberrant synaptogenesis and neuroinflammation in the hippocampus but not with mitochondrial dysfunction and oxidative stress, possibly due to the results of locally increased levels of progesterone metabolites. This would be an important study focusing on the female brain, which is often neglected in animal experiments. There are, however, some issues to be addressed to further improve the manuscript.

1.Why did the authors just focus on T1DM, not on T2DM? Because more than 90% of DM is type 2, the data of T2DM should be more useful for clinical consideration. Although streptozocin (STZ)-treated model is very popular, the reliable rat model of T2DM should be commercially available.

Answer: We agree with the Reviewer that T2DM is more prevalent on the population. However, to evaluate memory (or cognitive) problems in T1DM is relevant. Indeed, when assessed for dementia, patients with type 1 DM have a 65% increased risk of dementia, compared to the 37% of type 2 DM patients (see for instance Smolina et al., 2015 doi: 10.1007/s00125-015-3515-x). Anyway, we are also currently performing experiment in an animal model of T2DM that will be material for a future publication.

2. As to the evaluation of aberrant synaptogenesis, the authors only analyzed the expression level of synaptic proteins such as synaptophysin, synapsin and syntaxin. The morphological changes of synapse should be required.

Answer: Evaluation of the synaptic morphology is an interesting point. Unfortunately, we did not prepare tissues for histochemical evaluation. In addition, as already replied to the other Reviewer, the authorization to perform experiments on this experimental model expired, so at this time we cannot perform this kind of analysis. Anyway, morphological analysis is an important aspect that we wish to evaluate in our future experiments.

3. Although the authors used whole hippocampus and cerebral cortex, it is too rough. For the hippocampus, it should be dorsal or ventral, and for the cortex, the region should be more restricted.

Answer: We agree with the Reviewer that different sub-regions of hippocampus (ventral or dorsal) might present different features. Moreover, we know that prefrontal and perirhinal cortex are also relevant for the NOR test. However, our main target was to explore the role of neuroactive steroids in impaired memory and related molecular alterations occurring in the hippocampus and cerebral cortex. Indeed, even if liquid chromatography tandem mass spectrometry is a sensitive technique to evaluate neuroactive steroid levels, this requires amount of tissues greater than the single sub-regions mentioned above. Thus, to correlate data about steroid levels and molecular findings we were forced to use the entire brain areas.

Round 2

Reviewer 2 Report

It is necessary to properly state in the manuscript why only type 1 diabetes was focused instead of type 2 diabetes.

Author Response

We have now added this aspect in the introduction and conclusions (please see in red the modifications)